# Sensitivity of a 29-Channel MEG Source Montage

**DOI:** 10.3390/brainsci12010105

**Published:** 2022-01-13

**Authors:** Jukka Nenonen, Liisa Helle, Amit Jaiswal, Elizabeth Bock, Nicole Ille, Harald Bornfleth

**Affiliations:** 1Megin Oy, Keilasatama 5, FI-02150 Espoo, Finland; liisa.helle@megin.fi (L.H.); amit.jaiswal@megin.fi (A.J.); elizabeth.bock@megin.fi (E.B.); 2Department of Neuroscience and Biomedical Engineering, School of Science, Aalto University, FI-00076 Aalto, Finland; 3BESA GmbH, 82166 Gräfelfing, Germany; nille@besa.de (N.I.); harald.bornfleth@besa.de (H.B.)

**Keywords:** magnetoencephalography, source montage, epilepsy data review

## Abstract

In this paper, we study the performance of a source montage corresponding to 29 brain regions reconstructed from whole-head magnetoencephalographic (MEG) recordings, with the aim of facilitating the review of MEG data containing epileptiform discharges. Test data were obtained by superposing simulated signals from 100-nAm dipolar sources to a resting state MEG recording from a healthy subject. Simulated sources were placed systematically to different cortical locations for defining the optimal regularization for the source montage reconstruction and for assessing the detectability of the source activity from the 29-channel MEG source montage. The signal-to-noise ratio (SNR), computed for each source from the sensor-level and source-montage signals, was used as the evaluation parameter. Without regularization, the SNR from the simulated sources was larger in the sensor-level signals than in the source montage reconstructions. Setting the regularization to 2% increased the source montage SNR to the same level as the sensor-level SNR, improving the detectability of the simulated events from the source montage reconstruction. Sources producing a SNR of at least 15 dB were visually detectable from the source-montage signals. Such sources are located closer than about 75 mm from the MEG sensors, in practice covering all areas in the grey matter. The 29-channel source montage creates more focal signals compared to the sensor space and can significantly shorten the detection time of epileptiform MEG discharges for focus localization.

## 1. Introduction

Magnetoencephalography (MEG) is routinely used for recording epileptiform discharges and is used in many hospitals for the presurgical evaluation of patients with epilepsy [1,2,3,4,5,6]. MEG provides an accurate representation of the magnetic field distribution over the scalp. However, the high number of channels (306 in a MEGIN system) makes the manual search of epileptiform discharges time consuming as only a subset of channels can be viewed at once, and thus each MEG recording needs to be browsed through several times consecutively to review all sensors for potential epileptiform events. Therefore, source montages have been proposed for transforming the sensor-level MEG and electroencephalography (EEG) signals into virtual source-space signals representing the neural activity in different brain regions. Due to a substantially lower number of channels, the source montages facilitate much faster identification and evaluation of epileptiform activity than the sensor-level signals [6,7,8,9].

The estimation of the source-space signals requires solving an inverse problem that in a general case does not have a unique solution. A restricted inverse problem is, however, solvable, for example when the neural activity is estimated in terms of a limited number of current-dipole sources at pre-determined locations [9,10,11]. Source montages utilize a special spatial filter for converting the MEG and EEG sensor-level signals into the waveforms of standard regional sources in the cortex [8,9]. A regional source is fixed to the local brain structure by assuming one equivalent location in the depth of the gyrus or cortical sub-region, having two tangential components for MEG in a spherical head model, or three orthogonal components for the corresponding EEG data [8].

In the present study, we aim to systematically assess the reliability of the MEG source montage BR29 introduced in [6], representing 29 regions evenly distributed throughout the cortex. The source montage steps involve: (1) segment and co-register individual MRI and define a subject-specific spherical head model; (2) place regional sources at 29 standard locations on the cortex, having 2 orthogonal, tangential orientation vectors for MEG; (3) calculate the lead field vectors for the source dipoles; (4) create an inverse of the lead field matrix; and (5) store the inverse as source montage to be applied as a linear transform to the corresponding continuous MEG data. The amplitudes of these orthogonal dipoles vary over time. Instead of showing 2 × 29 MEG (or 3 × 29 EEG) waveforms, the regional source outputs can be presented as a single trace at each regional source location representing the maximum variance of the dipole activities. In this case, a regional source can be considered as an equivalent to a variable-orientation or rotating dipole [10,11], where the orientation is optimized for each individual analysis time window.

## 2. Materials and Methods

### 2.1. Spatial Filtering from Sensor to Source Space

Assume first that the source space consists of *P* current dipoles with fixed orientations. Let us denote by$\;b\left(t\right)$ an M×1 data vector of *M* measurement channels (MEG or EEG) for a single timepoint t, and by $s\left(t\right)$ as the corresponding P×1 vector in the source space for *P* dipoles. We express the relation between the samples in the sensor and source spaces as
(1)$$b\left(t\right)=L\;s\left(t\right)+n\left(t\right)$$
where L is the M×P lead field matrix whose column vectors define the contributions of the *P* dipoles at locations $r_{j}$ to the measurement $b\left(t\right)$, and $n\left(t\right)$ refers to measurement noise. The source amplitudes can be reconstructed with the regularized pseudoinverse as [8]:(2)$$\hat{s}\left(t\right)={\left(L^{T}L+\Lambda \right)}^{-1}L^{T}\;b\left(t\right)\;.$$

Here, Λ is a diagonal matrix of regularization parameters. The comparison between linear inversion approaches [12] reveals that the generalized linear solution for Equation (1) can be expressed as
(3)$$\hat{s}\left(t\right)=C_{x}L^{T}{\left(LC_{x}L^{T}+C_{n}\right)}^{-1}b\left(t\right)={\left(L^{T}C_{n}^{-1}L+C_{x}^{-1}\right)}^{-1}L^{T}C_{n}^{-1}b\left(t\right)\;.$$

The matrix $C_{x}$ represents some a priori information of the source covariances and $C_{n}$ is the noise covariance of the measurement. Equation (2) is obtained by setting the noise covariance to a unit matrix, and the regularization matrix Λ corresponds to $C_{x}^{-1}$ in Equation (3).

Because the MEG devices manufactured by MEGIN contain two kinds of sensors, gradiometers and magnetometers producing signals with different units, a scaling between the signals is needed when all sensors are used. One way is to perform noise whitening for the data and the lead field matrix [13] leading to
(4)$$\hat{s}\left(t\right)={\left(L^{T}C_{n}^{-1}L+\Lambda \right)}^{-1}L^{T}\;C_{n}^{-1}b\left(t\right).$$

We can apply either a full noise covariance if it is available or approximation with a diagonal noise covariance. In the latter case, the inverse matrix is $C_{n}^{-1}=diag\left(\sigma_{1}^{-2},\ldots ,\sigma_{M}^{-2}\right)$, where $\sigma_{i}$ represents the noise variance of the *i*th channel. Constant values of 50 fT/cm for gradiometers and 200 fT for magnetometers represent default noise variances in unaveraged epochs.

To reduce the depth bias, we rewrite Equation (1) as
(5)$$W_{b}\;b\left(t\right)=W_{b}\;L\left(W_{s}^{-1}W_{s}\right)\;s\left(t\right)+W_{b}\;n\left(t\right),$$
where $W_{b}=C_{n}^{-1/2}$, or $W_{b}=$ $diag\left(\sigma_{1}^{-1},\ldots ,\sigma_{M}^{-1}\right)$ for the diagonal case. The elements of the diagonal matrix $W_{s}$ contain the column norms of matrix $W_{b}\;L$. Denoting $b_{n}=W_{b}\;b\left(t\right),$ $L_{n}=W_{b}L\;W_{s}^{-1},$ and ${\hat{s}}_{n}=W_{s}\;\hat{s}\left(t\right),$ we can rewrite Equation (4) as
(6)$${\hat{s}}_{n}\left(t\right)={\left(L_{n}^{T}L_{n}+\Lambda \right)}^{-1}L_{n}^{T}\;b_{n}\left(t\right)=G^{-1}L_{n}^{T}\;b_{n}\left(t\right)$$
where we have denoted the regularized Gram matrix as $G=\left(L_{n}^{T}L_{n}+\lambda \;I\right)$. Because the diagonal values of $L_{n}^{T}L_{n}$ are equal to one, the regularization term can be written as Λ=λ I and the regularization parameter λ thus represents a percentage value.

### 2.2. Regional Sources

A regional source summarizes one location with all possible orientations (for MEG two tangentially oriented orthogonal orientations in the spherical volume conductor; for EEG three orthogonal orientations) of dipole sources at that location [8]. Three display modes are possible for regional sources:Individual amplitudes of the orthogonal dipole sources: 58 waveforms for MEG or 87 for EEG.Root mean square of the amplitudes of all dipolar components of a regional source: 29 waveforms.Amplitude of the principal component of the dipolar sources comprising the regional source: 29 waveforms.

In the last case, the orientations are obtained from
(7)$$\left({\hat{s}}_{j}^{T}{\hat{s}}_{j}\right)u_{k}=\lambda_{k}u_{k},\;$$
where ${\hat{s}}_{j}\;$ contains the waveforms of the *j*th regional source and $\lambda_{k}\;\mathrm{and}\;u_{k}$ are the eigenvalues and eigenvectors of the 2×2 matrix for MEG, or the 3×3 matrix for EEG, respectively. Then, the eigenvector $u_{k}\left(\theta ,\varphi \right)$ corresponding to the largest eigenvalue gives the best dipole orientation for the *j*th location during the analysis time window.

These options were first implemented in the commercial software packages EEGFocus and BESA Research (BESA GmbH, Gräfelfing, Germany; [7]). The first option gives the full information, but the number of waveforms is larger. The latter two options reduce the number of waveforms. The orientation optimization may be better for spike detection than the root mean square, but the orientations need to be optimized separately for each displayed data interval.

### 2.3. Simulated MEG Signals

MEG data with known current dipole position, amplitude, and orientation, as well as temporal waveforms, were simulated using a custom in-house software (MEGIN Oy). The simulated sources had a period of 200 ms of sinusoidal waveforms per second. These simulated dipoles were superposed with an existing spontaneous brain measurement to have realistic noise levels and background activity, and the total length of 10 s. MEG and EEG data were simulated using spherical or 3-layer BEM models, respectively. All the data were filtered to the frequency band 2–70 Hz.

Sensor-level MEG data were converted into BR29 montage source waveforms according to Equation (6) using in-house software (MEGIN Oy). The SNR for each dipole and reconstructed source montage amplitudes were estimated as well as highest sensor-level and source montage SNR. Diagonal noise covariance with the default noise variances was used for depth weighting in Equations (5) and (6). The simulation study was carried out in three phases.

**Simulation 1**: A pilot set with dipoles at the 29 source montage locations was simulated with 20-Hz sinusoidal dipole waveforms. These simulations were used to verify that the simulations produce suitable data with a physiological dipole amplitude of 100 nAm. By placing a tangential dipole at each source montage locations, an optimal regularization parameter was estimated.

**Simulation 2**: Another set of 50 tangential dipoles at various cortical locations in the left and right hemispheres was utilized to determine the detectability of the signals from the BR29 source montage. The dipole amplitude was again 100 nAm.

**Simulation 3**: In the third phase of the simulations, 4098 oblique cortical dipoles per hemisphere were used for a systematic seeding of source dipoles for the simulations. Additionally, here the dipole amplitude was 100 nAm. SNRs were computed for the source montage locations and the results were plotted as a map for the input source locations covering the cortex [14].

The 29 source montage locations are individually transformed into the subject-specific source space based on segmented MRI data. The geometry for the simulations was obtained from the segmentation of the MRI data of a healthy volunteer (60-year-old male) using the FreeSurfer software [15,16]. A subset of the montage locations is displayed in Figure 1.

### 2.4. Signal-to-Noise Ratio and Source Detectability

The SNR was estimated in dB from the ratio between the signal power in the signal time window ($s_{1}-s_{2}$) divided by the signal power in the baseline window ($r_{1}-r_{2}$):(8)$$SNR_{k}=20\;\log\left[\;\sum_{t_{i}=s_{1}}^{t_{i}=s_{2}}\frac{1}{s_{2}-s_{1}}{\left(b_{k}\left(t_{i}\right)-ave_{k}\right)}^{2}/\sum_{t_{i}=r_{1}}^{t_{i}=r_{2}}\frac{1}{r_{2}-r_{1}}{\left(b_{k}\left(t_{i}\right)-ave_{k}\right)}^{2}\right],$$
where $ave_{k}$ is the mean signal value of channel k over the baseline. The SNRs were determined for all MEG and source montage channels during each 1000 ms epoch (800 ms for the baseline and 200 ms for the signal), and the values were averaged over 10 epochs.

Corresponding SNR estimates can be computed from the source montage signals by replacing $b_{k}\left(t_{i}\right)$ in Equation (8) by ${\hat{s}}_{n,k}\left(t_{i}\right)$ of Equation (6) for the *k*th source montage channel. In the following, we denote the sensor and source level SNRs as SNR1 and SNR2, respectively. Among the SNRs of all channels, the highest SNR1 and SNR2 values were used for the analysis.

The estimated visual detectability SNR2 limit for the simulated signals is about 15 dB (see the Results section). For assessing the detectability of each simulated source, we used the count *Ndt* expressing how many source montage channels have the SNR of 15 dB or better. A smaller value of *Ndt* indicates more focal activity of the simulated signal, whereas value of *Ndt* = 0 naturally indicates that the simulated activity is not detectable in the source montage. Below, we indicate the sensor-level and source montage counts by *Ndt1* and *Ndt2*, correspondingly.

## 3. Results

### 3.1. Source Dipoles at Montage Channel Locations

Simulation 1 was performed by setting a tangential dipole to one of the 29 source montage locations at a time. Each dipole was excited with four cycles of 20 Hz sinusoidal signal (epoch duration 200 ms) repeated once per second. The dipole moment was $Q=100\;\mathrm{nAm}*\left(e_{\theta }+e_{\varphi }\right)$, where $e_{\theta },e_{\varphi }$ are the two tangential spherical coordinates unit vectors.

A regularization parameter was first determined for Equation (6) (Λ=λ I). The regularization determines the condition number (the ratio between the largest and smallest eigenvalue) of matrix G. The regularization parameter was studied separately for using all 306 MEG channels, as well as when using only 204 planar gradiometers or 102 magnetometers. Appendix A Figure A1 presents the details. Figure 2 shows the source montage waveforms for different regularization parameters estimated from signals originating from the red dipole in Figure 1.

Typically, regularization decreases the noise level in the data, which is also clear from Table 1 that summarizes the SNR results for the source dipoles at the 29 sources montage locations. In addition to the SNR values, we also listed the amplitudes (nAm) from the source montage channels with the highest SNR (Table 2). For the SNR, increasing the regularization improves the result, and the SNR of source montage channels reaches that of the sensor channels once regularization is increased to 2%. Increasing the regularization further does not improve the source-montage SNR. The expected amplitude for the montage channel of the source location is 100 nAm, but increasing the regularization value decreases the reconstructed source amplitude (Table 2). The regularization of 2% provides a good trade-off between SNR and amplitude conservation. This regularization is used in the remainder of this paper.

### 3.2. Source Dipoles at 50 Cortical Locations

In Simulation 2, we utilized the same simulation dipole locations as in [13]. Tangential dipoles $Q=100\;\mathrm{nAm}*\left(e_{\theta }+e_{\varphi }\right)$ were placed one by one at 25 locations in the left and another 25 related locations in the right hemisphere (Figure 3).

The SNR results for the source dipoles estimated from 306-channel data are summarized in Table 3 and Figure 4 (SNR1 for the sensor-level and SNR2 for the source-level SNR). Visual observations of the BR29 signals revealed that the source signals are detectable when the SNR2 is above 15 dB, corresponding to sources that are closer than approximately 75 mm from the sensors (Figure 4a). Table 3 lists also the mean, median and maximum number of montage channels in which SNR is at least 15 dB (*Ndt1*, *Ndt2*). The mean, median and maximum values of the distance from the sources to the nearest sensor were, correspondingly, 63, 54 and 107 mm. The corresponding values of the distance from the source to the montage channel with highest SNR were 33, 27 and 95 mm.

Figure 5 shows an example of sensor-level and source montage signals from a source at the left temporal cortex. Figure 6, in turn, presents an example of simulations where the source was close to the limit of detectability on the source montage signals. Figure 7 shows the isocontour maps of the signals in Figure 5 and Figure 6, and the locations of the deepest sources that are barely or not distinguishable from the sensor-level or source-montage signals.

### 3.3. Source Dipoles on Cortical Surfaces

In Simulation 3, we examined systematically cortical source locations of the left and right hemispheres. Each cortical dipole was oriented along the normal vector for the segmented pial surface following the boundary between white and grey matter. Thus, the simulated sources consisted of a dipole at each cortical surface node and its nearest neighbors, corresponding to a cortical patch of approximately 1–2 cm^2^. Statistics of the sensor-level SNR, source-montage SNR and source-montage amplitudes are summarized in Table 4 and Table 5.

The head position in the simulations was taken from a real recording and the right hemisphere was slightly closer to the sensors than the left one, which is reflected by the higher values for the right hemisphere in Table 4 and Table 5. Figure 8 was plotted on the cortical surface reconstruction using MATLAB^®^ to show the distribution of SNRs for sensor-level signals (SNR1) and source-montage signals (SNR2) with regularization levels of 0% and 2%. The color map indicates a higher detectability of the sources in the neocortex than in deeper brain structures. As is apparent from Figure 8, simulated source locations included lateral as well as medial surfaces, including the corpus callosum.

## 4. Discussion

Source montages provide a practical approach to review continuous MEG and EEG data and detect epileptiform discharges without swapping between different channel selections. Generally, the MEG source montage BR29 produced good detectability with most of the simulated sources, except for purely radially oriented sources or the sources at deepest locations in the brain, which are in any case at lowest signal amplitude on the sensor level [17]. The SNR values were evaluated and compared between sensors and source-montage channels. Generally, the SNR of source montages depends on regularization; without regularization, sensor-level signals showed higher SNR than the source-montage channels and higher *Ndt* counts (number of channels with SNR >= 15 dB). When using adapted regularization of 2%, the SNR of source montages increased to values comparable with the sensor-level values and the *Ndt2* counts became closer to the *Ndt1* values in Simulation 3 (Table 5). In particular, the regularization reduced the background noise levels in the source montage signals (as can be seen in Figure 5 and Figure 6).

Although epileptiform discharges have a sharper shape and shorter duration than the simulated sinusoidal source waveforms, our simulations demonstrate the performance of the source montages. The SNR definition in Equation (8) represents the mean power of the signal and baseline over the corresponding time windows used for the simulation data. Thus, similar SNR results can be obtained also when simulating shorter spike-like source signals with the same amplitude of 100 nAm and adjusting the signal time window accordingly. The longer signal window in our study was useful for reducing the effects of the variations in the underlying resting state date and provides clearer visualizations (Figure 2, Figure 5 and Figure 6). The detectable visual SNR that was estimated to 15 dB was based on the individual perception of the authors; it may be affected to some extent by the scenario of shorter signals and may also depend on the individual who assesses the signals, for both sensor space and source space signals. On the other hand, the amplitude of the epileptic spikes also varies and can be higher than that in the simulations.

The use of the source montage could reduce the time required for reviewing MEG data, approximately to 1/8 of the original. Another method for efficient data review is provided by the so-called butterfly plots, where the sensor-level MEG or EEG signals from the same anatomical area are superimposed on top of each other to make a simultaneous review of all the possible data. This approach, however, could disturb the detection of fine details in the data, if individual signals are not viewed. A different approach to reconstruct source-level signals from MEG or EEG recordings is obtained via the spatial filtering technique beamforming, where the so-called virtual electrodes can be estimated in the source space [18]. An advantage of the beamformer approach is the method’s ability to typically increase the source SNR compared to that of sensor data [19]. Although visual data review for the detection of epileptiform activity is still commonly used in the clinical work, with the emergence of reliable methods for automated spike detection [20,21,22,23,24,25,26,27,28], the use of visual data review might lose part of its relevance. At this stage, however, where the automated spike detection methods provide suggestions instead of ready solutions, the visual review of data remains an important step in epilepsy analysis.

Basically, similar source montage waveforms can be produced using either all 306 MEG channels or 204 gradiometers or 102 magnetometers separately, provided that the regularization is optimally set. The spatial filter matrix G in Equation (6) has the lowest condition number for the subset of 204 gradiometers and it is consequently more stable than the matrix for 102 magnetometers (see Appendix A Figure A1). As a result, reconstructions from the 306-channel mixed data and 204-channel gradiometer data produced a slightly better SNR and less noisy montage waveforms than the 102-channel magnetometer data. Furthermore, the reconstruction with 2% regularization produces slightly smaller source dipole amplitudes from the 102-channel magnetometer data than the reconstruction from the data with only gradiometers.

Reconstructed source montage amplitudes revealed that, on average, the montage channels see about 60 nAm amplitudes when the sources were in the set of 50 tangential dipoles with sinusoidal 100 nAm activity (Table 3), and about 30 nAm source amplitudes when cortical sources had a 100 nAm dipole with both radial and tangential components (Table 4).

The SNR distribution maps in Figure 8 indicate no significant blind areas on the cortex where the dipolar activity would remain undetected in the BR29 montage. The smallest SNR values (SNR2 < 15 dB) correspond to the deepest sources, which are over 75 mm from the nearest MEG sensor. As the MEG device configuration is such that in the optimal head position the source-to-sensor distance is at least approximately 30 mm, this would mean the most superficial ~45 mm from the scalp surface. Appendix A Figure A2 illustrates these brain regions.

Our simulations applied sinusoidal source waveforms, but the findings are also applicable to more complex epileptiform discharges. We focused on MEG data, but similar considerations with source montages can be also performed with EEG [7,8,9]. EEG electrodes are closer to the cortex than MEG sensors and EEG is more sensitive to radially oriented sources. Therefore, source montages computed from EEG are expected to provide an equally good or better detection of epileptiform discharges than BR29, provided that the number of EEG electrodes is sufficient for adequate spatial sampling over the scalp. Still, MEG is generally considered more accurate and reliable for source localization than EEG. In practice, the combination of MEG and EEG has provided the best detection of interictal discharges [2].

## 5. Conclusions

Our results demonstrate that the source montage signals computed from 306-channel MEG data are feasible in reviewing MEG signals due to higher signal focality in source-space and comparable SNR to sensor space. They can considerably shorten the amount of time needed for MEG evaluation of epileptiform signals and add additional information about the approximate location in source space.

## Figures and Tables

**Figure 1 brainsci-12-00105-f001:**
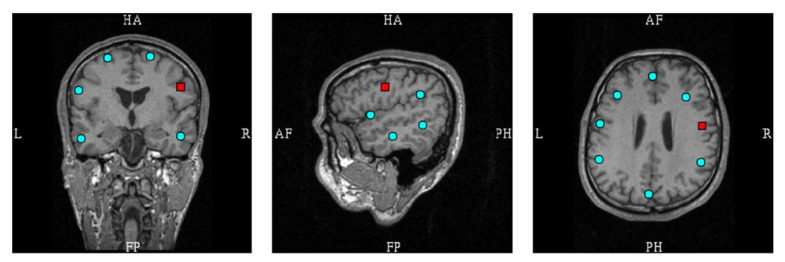
Subset of source montage dipoles overlaid on individual anatomical MR images. The red box indicates one location (FC6R) that was used for illustrative examples in Figure 2.

**Figure 2 brainsci-12-00105-f002:**
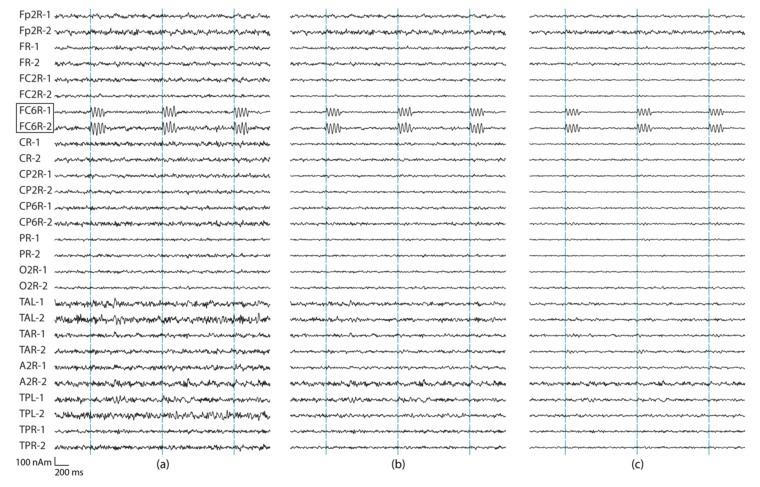
Source montage waveforms at a subset of 14 regional source locations estimated from all 306 channels. The simulated source was at the location FC6R. Three seconds data of the two tangential dipole orientations at each location (labeled as 1,2) are shown. Three different regularization values were used: (**a**) λ = 0%, (**b**) λ = 0.5%, and (**c**) λ = 2%.

**Figure 3 brainsci-12-00105-f003:**
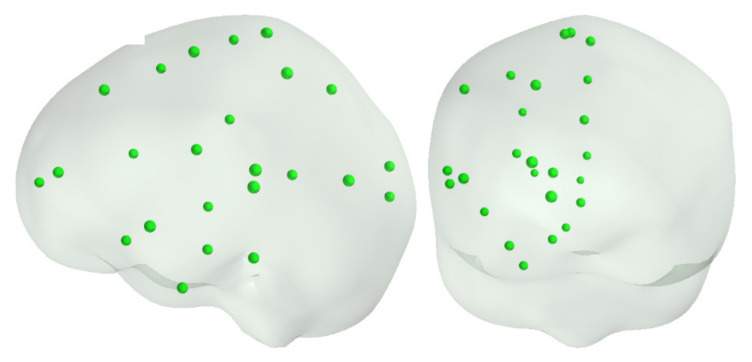
Locations of selected 25 sources used in [13], covering the left hemisphere. A corresponding set of points was also used on the right hemisphere.

**Figure 4 brainsci-12-00105-f004:**
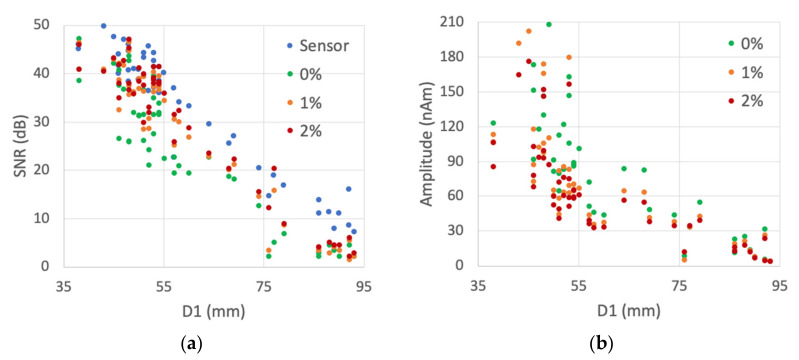
Simulation 2 results with three regularization values (0%, 1% and 2%). (**a**) Sensor-level SNR1 and source-level SNR2 distribution as a function of the distance D1 from the source to nearest sensor. (**b**) Source montage amplitude distribution as a function of D1.

**Figure 5 brainsci-12-00105-f005:**
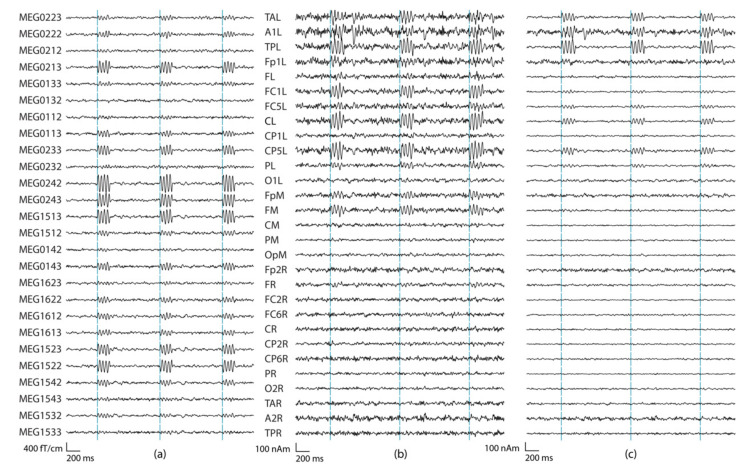
Examples of sensor-level and source-montages signals where the reconstructed activity is detectable at multiple sensor-level and source-montage channels. (**a**) Left temporal gradiometers. (**b**) Source montage waveforms at all 29 regional source locations of BR29 with no regularization, (**c**) and with 2% regularization. The source montage dipole orientations were optimized according to Equation (7). The source was a tangential dipole with the amplitude of 100 nAm in a left temporal location. The highest sensor SNR was 40 dB (MEG0242), and highest source montage SNR was 38 dB (TPL; distance to the nearest sensor is 48 mm, distance to the source montage channel with the highest SNR is 24 mm, Q = 99 nAm). In (**a**) *Ndt1* = 46, in (**b**) *Ndt2* = 6, and in (**c**) *Ndt2* = 8.

**Figure 6 brainsci-12-00105-f006:**
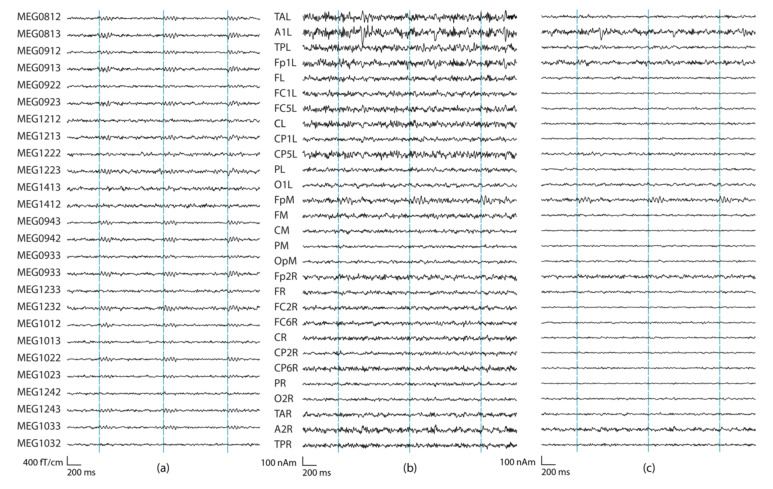
Examples of sensor-level and source-montage signals where the reconstructed activity is barely detectable. (**a**) Right frontal gradiometers. (**b**) BR29 montage waveforms with regularization 0%, (**c**) as in (b) but with 2% regularization. The source was a tangential frontal dipole with the amplitude of 100 nAm in a frontal location. Highest sensor SNR was 20 dB (MEG1022), and highest source montage SNR was 15 dB (FpM; distance to the nearest sensor is 74 mm, distance to the source montage channel with the highest SNR is 13 mm, Q = 35 nAm). In (**a**) *Ndt1* = 20, in (**b**) *Ndt* = 0, and in (**c**) *Ndt* = 1.

**Figure 7 brainsci-12-00105-f007:**
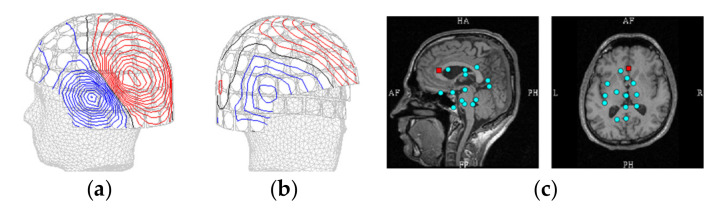
(**a**) Isocontour maps at peak signal of Figure 5a. (**b**) Corresponding map at peak signal of Figure 6a. (**c**) Locations of the 15 dipoles with lowest SNR; the red box points to the source dipole of Figure 6.

**Figure 8 brainsci-12-00105-f008:**
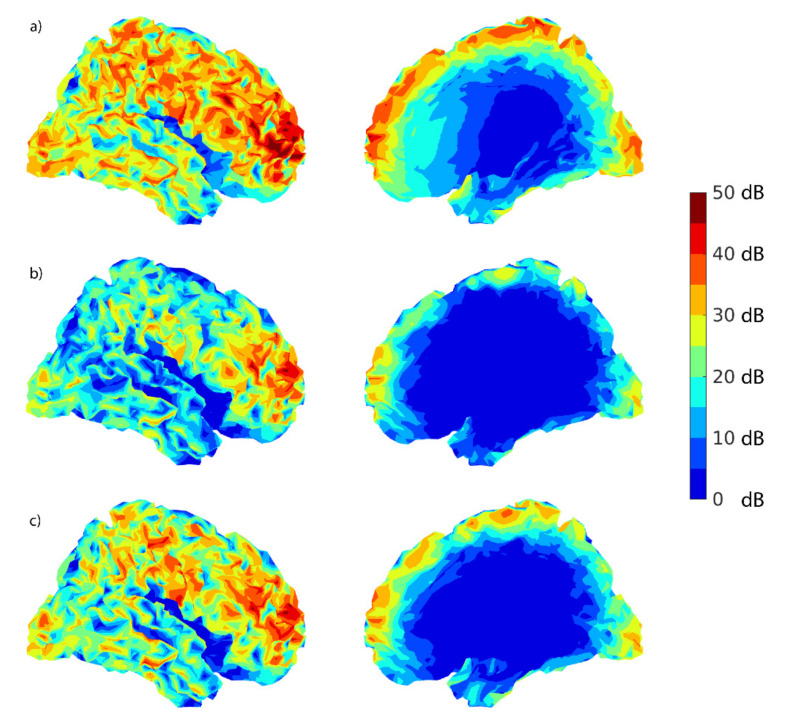
Lateral and sagittal view for (**a**) sensor-level SNR1 cortical maps (in dB) on the right hemisphere, (**b**) source-level SNR2 cortical maps (in dB), regularization 0%, and (**c**) source-level SNR2 cortical maps (in dB), regularization 2%. These plots represent the highest SNR values among all channels for each simulated cortical source location.

**Table 1 brainsci-12-00105-t001:** SNR values (dB) among sensors (SNR1) and source montage channels (SNR2) in Simulation 1. These values are averages of the highest SNRs for the 29 sources at the source montage locations.

	SNR1	SNR2 λ = 0%	SNR2 λ = 0.5%	SNR2 λ = 1%	SNR2 λ = 2%	SNR2 λ = 3%	SNR2 λ = 4%	SNR2 λ = 5%
306 chns	37	28	33	35	37	38	38	38
204 grads	37	29	33	35	37	38	38	39
102 mags	29	17	31	32	34	34	35	35

**Table 2 brainsci-12-00105-t002:** Amplitude values (nAm) of the source montage channels with the highest SNR in Simulation 1. These values are averages of the 29 sources at the source montage locations. The simulated source amplitude was 100 nAm.

	Λ = 0%	λ = 0.5%	λ = 1%	λ = 2%	λ=3%	λ=4%	λ = 5%
306 chns	100	86	80	73	69	66	63
204 grads	100	89	84	78	73	70	68
102 mags	94	71	64	58	54	51	49

**Table 3 brainsci-12-00105-t003:** Simulation 2: mean, median and maximum values of the sensor-level signal-to-noise ratio SNR1 (in dB), source-level signal-to-noise ratio SNR2 (in dB), number of source montage channels with SNR >= 15 dB (*Ndt1* from the sensor-level and *Ndt2* from the source montage signals), and the source amplitude Amp (in nAm) at the montage channel with highest SNR2. The results summarize the simulations of 306-channel MEG data from the 50 cortical dipole locations with three regularization values. Cases with *Ndt* = 0 (SNR < 15 dB) were also included in computing the mean *Ndt* values.

	SNR1	*Ndt1*		SNR2			*Ndt2*			Amp	
λ	-	-	0%	1%	2%	0%	1%	2%	0%	1%	2%
mean	31	33	23	26	28	4.4	3.9	3.8	94	67	60
median	38	44	25	32	34	2	3	4	84	64	56
max	50	71	47	47	47	19	14	12	263	202	176

**Table 4 brainsci-12-00105-t004:** Simulation 3: mean, median and maximum values of the sensor-level SNR1 (dB), source-level SNR2 (dB), number of montage channels with SNR >= 15 dB (*Ndt1* from the sensor-level and *Ndt2* from the source montage signals), and the source amplitude Amp (nAm) at the montage channel with highest SNR2. Simulated source amplitude was 100 nAm. Cases with *Ndt* = 0 (SNR < 15 dB) were also included in computing the mean *Ndt* values.

		SNR1	*Ndt1*	SNR2			*Ndt2*			Amp		
	λ	-	-	0%	1%	2%	0%	1%	2%	0%	1%	2%
Left	mean	21	11	10	14	15	0.6	1	1.2	41	30	28
	median	21	5	7	13	15	0	0	1	31	25	23
	max	52	55	44	48	48	12	10	9	257	191	166
Right	mean	22	12	13	16	18	1.3	1.5	1.7	42	32	29
	median	24	8	12	16	18	0	1	1	31	25	23
	max	53	58	48	47	47	22	13	13	295	241	221

**Table 5 brainsci-12-00105-t005:** Simulation 3: the number of source patches that produced signals that were detectable at least on one source montage channel (*Ndt1* > 0 or *Ndt2* > 0, SNR >= 15 dB). The total number of simulated patches per hemisphere was 4098.

	Sensors	BR29		
	**-**	λ=0%	λ=1%	λ=2%
Left	2715	1182	1894	2059
Right	2840	1710	2186	2333

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
