# Peer review of "Sensitivity of a 29-Channel MEG Source Montage"

_brainsci, 2022, doi:10.3390/brainsci12010105_

Round 1
Reviewer 1 Report
In this paper, the authors indicated the detectability of epileptic spikes when their new idea is applied to the visual data review for the detection of epileptiform activity. The paper indicated the way to estimate SNR of reconstructed-source-based waveforms by several simulations. It will be interesting not only for reviewers of the MEG data containing epileptiform spikes, and worthy to be published. However, a minor revision is necessary in accordance with the following points to be modified or added. 
1. First of all, "source montage" is not a general term yet. A brief explanation for this term is required in Introduction. The 29 brain regions (BR29) also need to be explained briefly in Introduction even though the detail explanation is described in ref [5]. 
2. On L. 108, the meaning of "but it needs to be..." is ambiguous. Explain what it meant more clearly.
3. On L. 129-135, how much is the dipole amplitude for simulation 2 and 3? Still 100 nAm? 
4. On L. 157, the author suggested that a smaller Ndt indicates more focal activity of the simulated signal. However, if the Ndt were zero, it would mean the detection of the epileptic discharge failed but would decreased the mean and median of Ndt shown in Table 3 and 4. The author should discuss how the case of Ndt = 0 was treated.
5. Waveforms of sources were labeled differently between Fig. 2 and Fig. 5bc/6bc. These should be labeled in the same way if they were from the same regions. 
6. On L. 194, the unit for Q is nAm? 
7. In the caption of the figure 5 and 6, if the authors indicate the Ndt for (b) and (c) as examples, it will be more understandable how the Ndt is counted.
8. The units should be indicated for the color bar in Figure 8. The labels for vertical and horizontal axes in Figure A1 are also required.
9. On L. 281, the authors described the two lowest channels (A1L, A2R) have higher noise levels. But it seems that the noise levels of A1L and A2R are not so distinct among 29 channels in Fig. 5 and 6. Where to see for the larger noise? 
10. An epileptiform dischage is expected to have shorter duration and thus smaller power than sinusoidal signals applied to the simulations. Therefore, the authors should discuss if the detactability criteria of 15 dB of SNR is sufficient when real epileptiform spikes are considered in Discussion. 
11. On L. 356, is "all 3096 channels" typo for "all 306 channels"?

Author Response
We thank the reviewer for the encouraging response. We have made corrections to the manuscript and indicate the detailed changes under each item below.

  1. We elaborated the source montage description and modified the text on lines 35-45. We also swapped the order of refs 5-6 so that now refs 6-9 point to source montage studies.  We added more detailed description of the BR29 source montage (ref [6]) on lines 50-56.
  2. Reformulated the sentence slightly, now on line 121: “…but the orientations need to be optimized separately for each displayed data interval.”
  3. We added note that the dipole amplitude in Simulations 2 and 3 was 100 nAm.
  4. We added explanations,
  • Lines 208-210: “Value Ndt = 0 naturally indicates that the simulated activity is not detectable in the source montage.”
  • Lines 271-272 (Table 3) and 326 (Table 4): “Cases with Ndt = 0 (SNR2 < 15 dB) were also included in computing the mean values.”

We also added comparison between Ndt1 from the sensor-level signals (SNR1 >= 15 dB) and Ndt2 from the source-montage signals (SNR2 >= 15 dB); line 175 and Tables 3-5.
5.We elaborated the figure captions. Fig. 2 shows only a subset 14/29 regional sources but both dipole orientations at each location (labels 1,2 in channel names). Figs 5 and 6 show all 29 waveforms after the dipole orientations have been optimized.
6.Line 213: Added the unit Q = 100 nAm * (e_theta + e_phi)
7.We added the counts; Ndt1 from the sensor-level Ndt2 from the source-montage signals. Figure (5): In (a) Ndt1 = 46, in (b) Ndt2 = 6, and in (c) Ndt2 = 8. Figure (6): In (a) Ndt1 = 20, in (b) Ndt = 0, and in (c) Ndt = 1.
8.We added the labels in Figures 8 and A1.
9.This is not so clear in all cases, so we rewrote the sentence (lines 305-306): “In particular, the regularization reduced the background noise levels in the source montage signals (as can be seen in Figures 5 and 6).”
10.We added a paragraph in Discussion (lines 313-325): “Although epileptiform discharges have a sharper shape and shorter duration than the simulated sinusoidal source waveforms, our simulations demonstrate the performance of the source montages. The SNR definition in Eq. (8) represents the mean power of the signal and baseline over the corresponding time windows used for the simulation data. Thus, similar SNR results can be obtained also when simulating shorter spike-like source signals with the same amplitude of 100 nAm and adjusting the signal time window accordingly. The longer signal window in our study was useful for reducing the effects of the variations in the underlying resting state date and provides clearer visualizations (Figs. 2, 5, 6). The detectable visual SNR that was estimated to 15 dB was based on individual perception of the authors; it may be affected to some extent by the scenario of shorter signals and may also depend on the individual who assesses the signals, for both sensor space and source space signals. On the other hand, amplitude of the epileptic spikes also varies and can be higher than that in the simulations.”
11.Typo, corrected it to 306.

Reviewer 2 Report
In the paper ‘Sensitivity of a 29-channel MEG source montage’ the authors use simulated data to investigate the reliability of a 29-chennel MEG source montage to facilitate visual inspection of MEG data when looking for epileptiform spikes.
Despite the increasing development of automatic tools for spike detection, I think this paper may be of interest for the neuroscientific community as visual inspection of the data is still largely employed in clinical scenarios and it remains a key step to validate newly developed automatic tools.
However, I think some part of the results section should be better clarified before the paper being accepted for publication in brain sciences.
Below few issues I had in understanding the results presented by the authors.

Major issues:

  1. Section 2.4. It is not clear to me how the SNR at the source-montage level (SNR2) is computed.
    Indeed, in equation (8), b_k seems to be the recorded magnetic field at the k-th sensor. To compute SNR2 do the authors solve the MEG forward problem to compute the magnetic field b_k starting from the reconstructed source activity \hat{s}? Or do they define SNR2 with a formula similar to (8) but with the magnetic field b_k replaced by the reconstructed activity \hat{s}?
  2. Figure 2. I think it would be useful to add the reconstructed time courses at the location of the channel that is truly active in the considered simulation. If they are already present (was the dipole located in FC6R?) I would highlight such time-courses.
  3. Figure 8. According to equation (8) for each source location multiple values of SNR are computed (one for each MEG sensor/source-channel). What do the authors plot in Figure 8? The average SNR across all sensors? The highest value of SNR?
  4. Line 281-282 of the discussion. The authors wrote ‘except that two lowest channels (A1L, A2R, see Figure1) have higher noise values which slightly affect the SNR estimates’. I was not able to find any reference to this fact in the result section. Figure 1 seems to show only the location of A1L and A2R.

Minor issues:

  1. Second line of section 2.1. As far as I understood, the authors solve the MEG inverse problems by using tangential dipolar sources. If this is the case, I think the size of s(t) should be 2P x 1 (and 3P x 1 for EEG where free orientation is employed).
  2. Equation (1). I think the authors should explain the relationship between L and L(r_j) as it may be not clear to the unexpert reader. 
  3. Right hand side of equation (3). The reconstructed sources should depend on time t. 
  4. Line 101, the matrix \hat{s}_j^t {s}_j in eq. (7) should be 3x3 when EEG data are considered.
  5. I think the title of section 2.1. (‘Linear transformation’) and 3.3. (‘Cortical surfaces’) are a bit too generic.

Typos:
Line 115. I think the word ‘Both’ should be removed
Line 287-288. Detection of details in of the data → Detection of details in the data
Line 295. Might lose its part of its → Might lose part of its 

Author Response
We thank the reviewer for the encouraging response. We have made corrections to the manuscript and indicate the detailed changes under each item below.

Major issues:
1.We elaborated the description and added a paragraph on lines 165-169: “Corresponding SNR estimates can be computed from the source montage signals by replacing  in Eq. (8) by  of Eq. (6) for the kth source montage channel. In the following we denote the sensor and source level SNRs as SNR1 and SNR2, respectively. Among the SNRs of all channels, the highest SNR1 and SNR2 values were used for the analysis.”
2.We emphasized labels FC6R-1 and FC6R-2 as red in Figure 2 and added in the caption: “The simulated source was at the location FC6R.”
3.We added explanation on lines 168-169: ”Among the SNRs of all channels, the highest SNR1 and SNR2 values were used for the analysis.”; and in Figure (8) caption, lines 310-311: “These plots represent the highest SNR values among all channels for each simulated cortical source location.”
4.This is not so clear in all cases, so we rewrote the sentence (lines 305-306): “In particular, the regularization reduced the background noise levels in the source montage signals (as can be seen in Figures 5 and 6).”

Minor issues:
1.This is correct, but for kept the presentation more generic by added an explanation on line 64: “Assume first that the source space consists of P current dipoles with fixed orientations.”
2.We reformulated the text slightly on lines 70-71: “where L is the MxP  leadfield matrix whose column vectors define the contributions of the P dipoles at locations  r_j…”
3.Time dependency (t) is now on both sides of Eq. (3).
4.We added on line 114: “…of the  matrix for MEG, or the  matrix for EEG.”
5.We changed the section titles as 2.1. Spatial filtering from sensor to source space and 3.3 Source dipoles on cortical surfaces.

Typos:
Now line 128, removed ”Both”.
Now line 329, corrected to ”Detection of fine details in the data”.
Now line 337, corrected to ” Might lose part of its ”.

Round 2
Reviewer 2 Report
I thank the authors for kindly answering all my concerns. 
I think the paper is now suitable for publication in Brain Science.